# Matrix Metalloproteinases in Oral Health—Special Attention on MMP-8

**DOI:** 10.3390/biomedicines11061514

**Published:** 2023-05-23

**Authors:** Tsvetelina Atanasova, Teodora Stankova, Anelia Bivolarska, Tatyana Vlaykova

**Affiliations:** 1Faculty of Dental Medicine, Medical University of Plovdiv, 4002 Plovdiv, Bulgaria; fdm5475@mu-plovdiv.bg; 2Department of Medical Biochemistry, Faculty of Pharmacy, Medical University of Plovdiv, 4002 Plovdiv, Bulgaria; teodora.stankova@mu-plovdiv.bg (T.S.); anelia.bivolarska@mu-plovdiv.bg (A.B.); 3Department of Medical Chemistry and Biochemistry, Medical Faculty, Trakia University, 6000 Stara Zagora, Bulgaria

**Keywords:** matrix metalloproteinase, MMP-8, implant loss, peri-implantitis, periodontitis, caries

## Abstract

Matrix metalloproteinases (MMPs) are a large family of Ca^2+^ and Zn^2+^ dependent proteolytic enzymes, able to cleave the various components of the extracellular matrix (ECM), as well as a range of other regulatory molecules. Several reports have proven the important role of both MMPs and their endogenous inhibitors, TIPMs, in oral health, the initial development of the tooth, and during enamel maturation. In this mini-review, we aim to summarize the literature information about the functions of MMPs, paying more attention to MMP-8 (collagenase-2 or neutrophil collagenase) in the development and progression of periodontitis, peri-implantitis, and carious lesions. We also emphasize the role of particular gene variants in *MMP8* as predisposing factors for some oral diseases.

## 1. Introduction

Matrix metalloproteinases (MMPs) are a type of Ca^2+^- and Zn^2+^-dependent proteolytic enzymes. They are also called “matrixins”, and their main role is to cleave the various components of the extracellular matrix (ECM), as well as a range of other molecules, such as growth factors, cytokines, chemokines, and adhesion proteins [1]. In humans, the MMP family consists of more than 23 members, divided into six groups: collagenases, gelatinases, matrilysins, stromelysins, membrane-type MMPs, and other MMPs [2].

A variety of cell types are able to express and secrete MMPs as the main producers are the activated neutrophils, macrophages, endothelial cells, epithelial cells, vascular smooth muscle cells, glial cells, tumor cells, etc. [1].

Usually, after the activation of the cells, MMPs are released together with regulatory molecules such as interleukins (IL-8 and IL-1β), tumor necrosis factor (TNF)-α, osteoprotegerin (OPG), prostaglandins (PGE2), and receptor activator of nuclear factor kappa-B ligand (RANKL) [3].

The MMPs are involved in a vast range of physiological processes: angiogenesis, apoptosis, cellular differentiation, embryogenesis and morphogenesis, wound healing, immune responses, etc. However, the MMPs participate practically in all types of pathological conditions; for example, they are secreted and activated in diseases such as periodontal disease, rheumatoid arthritis, asthma, liver fibrosis, autoimmune diseases, and cancer [4].

MMPs are synthesized in non-active form as proenzymes (zymogenes, proMMP), and the MMP activity is regulated via gene expression, proenzyme activation, and endogenous tissue inhibitors, called tissue inhibitors of matrix metalloproteinases (TIMPs). There are four types of TIMPs that are present in humans (TIMPs 1–4). Together with MMPs, TIPMs play a major role in the remodeling of ECM and in the replenishing of its components [5,6].

Several reports have proven the important role of both MMPs and TIPMs in oral health, describing the implication of MMPs/TIPMPs balance in the initial development of the tooth and during the enamel maturation. They are also found in both intact and carious dentin, as well as in the pulp and in the saliva [7]. MMPs are expressed by different types of cells in the oral cavity. For example, in an early report from 1994, MMP-9 was mainly detected in gingival keratinocytes, while MMP-2 was expressed by gingival and granulation tissue fibroblasts [8]. Later, Tervahartiala et al., have described expressions of MMP-2, -7, -8, and -13 in gingival sulcular epithelium. In addition, MMP-7 and -13 were also found to be secreted by fibroblasts and macrophages and MMP-8 by neutrophils [9].

In this mini-review, we aim to summarize the literature information about the functions of MMPs, paying more attention to MMP-8 (collagenase-2 or neutrophil collagenase) in the development and progression of periodontitis, peri-implantitis, and carious lesions. We also emphasize the role of particular gene variants in *MMP8* as predisposing factors for some oral diseases.

The review was done on the bases of international databases, such as PubMed (National Library of Medicine, Bethesda, MD, USA), Google Scholar, EBSCO (EBSCO Industries, Ipswich, MA, USA), Scopus (Elsevier, Amsterdam, The Netherlands), and Web of Science (Clarivate Analytics PLC, London, UK). We performed a search using keywords MMPs, metalloproteinase, MMP-8, *MMP8*, *MMP8* polymorphism, oral health, periodontitis, peri-implantitis, periimplantitis, implant loss, caries. We summarized the results and conclusions from the papers. The papers included in the review are mainly from the period of 2010–2022.

## 2. MMP-8

Matrix metalloproteinase 8 is a type of collagenase (collagenase-2 or neutrophil collagenase). It is a glycoprotein and as all MMPs, it is synthesized as a zymogene (proMMP-8), which during the activation, undergoes proteolytic cleavage, leading to the removal of 80 amino acid propeptide from the N-terminus [10]. MMP-8 is released mostly by polymorphonuclear cells (PMNs, neutrophils), which store the MMP-8 pro-enzyme in specific granules, but it can also be expressed by various other cell types such as macrophages, T-cells, plasma cells, endothelial cells, vascular smooth muscle cells, fibroblasts, chondrocytes, keratinocytes, epithelial cells of bronchi, cornea, colon, gingival sulcus, etc. [6]. The main substrates of MMP-8 are ECM proteins as fibril collagens (types I, II, III), non-fibril collages (type IX, XII, XIV), fibronectin, laminin, entactin, tenascin C, and the proteoglycan aggrecan. MMP-8 is also able to cleave some non-matrix proteins and peptides such as angiotensin l, serpins, bradykinin, substance P, fibrinogen, α2-macroglobulin, bradykinin, α1-antitrypsin, CXCL5 (chemokine (C-X-C motif) ligand), IL-8 [2,10,11].

ProMMP-8 is released from PMNs, and then it is activated in the extracellular space using the cysteine switch mechanism [11]. In the gingiva, when secreting MMP-8, the activated immune and non-immune cells also release different cytokines and regulatory molecules (IL-8, OPG, PGE2, IL1β, TNF-α and RANKL) (Figure 1) [3,12]. One of the biological roles of MMP-8 is to facilitate the migration of neutrophil granulocytes from circulation into the tissues, including periodontium, by cleavage of collagen and other ECM components [13]. The uncontrolled increased expression, release, and activation of MMPs and other proteinases, including MMP-8, is thought to induce an inflammatory response, which leads to the destruction of periodontal tissues and other inflammatory diseases [14].

The *MMP8* gene is located in chromosome 11q22.3; in the same cluster as eight other MMP genes encoding MMP-1, MMP-13, MMP-3, MMP-10, MMP-7, MMP-12, MMP-20, and MMP-27 [15,16]. A number of gene variants have been identified in the promoter and in the encoding regions of the *MMP8* gene, as several of them have been proven to have allele-specific effects on the expression and/or enzyme activity [17,18,19]. Such naturally occurring sequence variations with functional activity are *MMP8* + 17 C > G (rs2155052), *MMP8* −799 C > T (rs1320632), −381 A > G (rs11225395) [17,18], −1089A > G (rs17099452), −815G > T (rs17099451), −795C > T (G > A) (rs11225395), −763A > T (rs35308160); [19], rs1940475 (C > T), rs3765620 (A > G) [20], and rs2508383 [21] (Figure 2).

Particularly, the *T* allele of the promoter polymorphism *MMP8* −795 C > T (G > A) (rs11225395) is bound with higher affinity by nuclear proteins leading to increase protein MMP-8 expression [23,24]. In chorion-like trophoblast cells, the promoter sequence containing the minor alleles of three single-nucleotide polymorphisms (SNPs) (−799C > T, −381A > G, and +17C > G) has been shown to have 2-3-fold higher activity in comparison to the major allele promoter construct (*C/A/C*) and to any other haplotypes with at least one major allele. However, in other cell types, such as U937 leukocyte cells or THP-1 monocytes/macrophage cells, this difference in promoter activity has not been detected, suggesting cell host dependence in *MMP8* promoter activity [10]. A recent study has reported that the haplotype, Hap3, constructed by the variant alleles *(GTTT*) of four other SNPs in the promoter of *MMP8* (−1089A > G [rs17099452], −815G > T [rs17099451], −795C > T or G > A, [rs11225395], −763A > T [rs35308160]) has lower promoter activity than the wild-type Hap1 haplotype (*AGCG*). The latter finding was explained by the diminished binding of the NF-kB to the −815T allele of Hap3 in comparison to the wild-type −815G allele [19].

The non-synonymous SNP in *MMP8*, the substitution of A with G at position 259 (+259A > G, rs. 1940475) leads to the replacement of Lysine 87 with Glutamate (Lys87Glu, K87E). This mutation is localized in the pro-domain of MMP-8 and results in the modification of the structural stability and, furthermore, the impairment of the catalytic activity of the enzyme [25].

In an extensive body of case-control studies and meta-analyses, the functional SNPs in *MMP8* have been shown to associate with the risk and progression of diseases such as colorectal and breast cancers [22,23,24], atherosclerosis [16], hypertension [19], bronchial asthma [26], osteoarthritis [27], and several oral pathological conditions such as periodontitis [5,17,18,28,29,30,31], gingivitis, caries, peri-implantitis and early implant failure [2,32].

## 3. MMPs and MMP-8 in Periodontal Diseases

Periodontitis is a multifactorial disease that causes soft tissue and bone loss. Severe periodontitis affects 740 million people worldwide and is the sixth most prevalent disease. The diagnosis is based on the evaluation of the standard clinical parameters [3].

The main initiator of this chronic inflammatory disease is the interaction between the pathogenic biofilm in the subgingival and the aberrant host immune response [33]. Evidence suggests that MMPs and their tissue inhibitors (TIMPs) play an important role in tissue remodeling and tissue destruction in general and in dental tissues as periodontal tissues in particular [7,34]. Periodontal inflammation is associated with the disruption of the balance between MMPs and TIMPs [35] (Figure 3).

Stimulation of the host cells by the pathogens from the dental plaque is considered a type of indirect mechanism of destruction of the tissues in periodontitis [5] (Figure 1).

Pathogens such as *Treponema denticola* (*T. denticola*), *Tannerella forsythia* (*T. forsythia*), and *Porphyromonas gingivalis* (*P. gingivalis*), which are the main components of pathogenic biofilm found in the gingival crevicular fluid and plague, induce a cascade which leads to the increased levels of an active form of several MMPs [4,36,37]. Several studies have demonstrated that those pathogens activate the secretion (e.g., MMP-2, MMP-9) and especially the activation of MMPs (e.g., MMP-8) both by bacterial-derived protease (a serine-type protease) and by the oxidative stress and release of myeloperoxidase (MPO) caused by the respiratory burst during the neutrophil phagocytosis [4].

In periodontal diseases, special attention has been paid to three collagenases (MMP-1, MMP-8, and MMP-13) and to the gelatinases (MMP-2 and MMP-9) because the major component of the ECM is collagen type I. All other MMPs (MMP-7, -12, -14) and proteases have relatively moderate effects in periodontitis [4].

Recently, MMP-8 has been considered to be one of the most promising biomarkers for early detection of periodontitis and for its progression and prognosis of treatment [3,4]. Elevated salivary and gingival crevicular fluid (GCF) MMP-8 levels have been reported in patients with initial and chronic periodontitis and in those with periodontitis linked to diabetes [3,38], while the antibiotic and/or scaling and root planning treatment, as well as the application of MMP inhibitory adjuvant medicines, have been shown to lead to the reduction of the level of MMP-8 [4,39,40,41]. For example, Yakov et al. reported that the persistent increase in MMP-8 in the gingival crevicular fluid is an indicator of a high risk of poor response to periodontal therapy [36]. Sensitive MMP-8-based assays of saliva, applied as chair-side kits, have been recently created [3]. These assays effectively distinguish clinically healthy sites and gingivitis from periodontitis and could also be applicable in monitoring the treatment of patients with chronic periodontitis [41]. Thus the introduction of that method for periodontal disease testing would give an advantage not only for the diagnosis but also for the identification of susceptible individuals and the prognosis of treatment [3,42].

Factors determining the susceptibility for periodontitis are some functional variants of the genes encoding the MMPs [17,43]. Variations in the *MMP8* gene that have been mostly investigated in association with periodontitis are *MMP8* −799 C > T (rs11225395) +17 C/G (rs2155052) and −381A/G (rs11225395) (Table 1).

As it is shown in Table 1, there are quite a few studies for *MMP8* gene variants in the development of periodontitis, and the results show that polymorphisms in the promoter regions (−799 C > T, −381 A > G) might be of importance as predisposing factors both for chronic and aggressive periodontitis [18,44]. The results of the two meta-analyses published lead to similar conclusions suggesting that the variant *T* allele of *MMP8* −799 C > T SNP is associated with an increased risk of periodontitis in four genetic models [29,43]. When the meta-analysis was performed in subgroups, it proved that the increased risk was mainly valid for Asians, for chronic periodontitis, and non-smokers [29].

There are also a few other MMPs evaluated in saliva and GCF and in periodontitis gingiva in patients with periodontitis. The immunohistochemically labeled cells for MMP-13 and for MMP-8 were higher in density in periodontitis gingiva when compared with healthy control tissue (*p* < 0.01). In periodontal diseases, gingival sulcular epithelium expresses several, rather than a single, collagenolytic MMPs, and this proteolytic cascade is evidently responsible for the tissue destruction characteristic of adult and juvenile periodontitis [9].

Besides MMP-8, decreased levels in gingival crevicular fluids after the effective treatment of periodontitis have also been observed for MMP-1, -9, -12, and -13 [4,46,47]. Indeed, the analyses of GCF from 29 African-American individuals diagnosed with localized aggressive periodontitis treated with full-mouth scaling and root planning and systemic antibiotics have proven that the levels of MMP-1, -8, -9, -12 and -13 were significantly reduced up to 6 months after the beginning of therapy and correlated positively with some clinical parameters as the pocket depth [46]. Even more, MMP-9 found in the saliva is shown to be a more sensitive biomarker during orthodontic treatment, which is promising for the decreasing of periodontal hazards during such manipulations [4].

In addition, significant associations were found between MMP-8 and MMP-9 activities in gingival crevicular fluid and the severity of the periodontal disease, together with negative correlations with TIMP-1 and TIMP-2 levels. This means that there is a dynamic in the balance between the active MMPs and their endogenous inhibitors, TIMPs, and suggests that MMP inhibitors could be a part of an innovative therapy against the effects of MMP on periodontal tissues [2]. Such MMP inhibitory effect is expressed by subantimicrobial doses of doxycycline, which have been approved as adjuvant therapy for treating periodontitis [48].

## 4. MMPs and MMP-8 in Peri-Implantitis

Implant-supported oral rehabilitation has attracted increasing attention due to high clinical success and proven improvement in patient quality of life [49]. However, it is associated with complications such as peri-implant mucositis (PIM) and peri-implantitis (PI). PIM is an inflammatory lesion confined to the soft tissues surrounding an endosseous implant in the absence of supporting bone loss or ongoing marginal bone loss [50]. Peri-implantitis is a localized infectious disease that causes an inflammatory response in both soft tissues and bone loss surrounding the osseointegrated implant. The microorganisms that are linked to implant failure are Gram-negative anaerobes and spirochetes. The traditional diagnosis is based on bleeding, changes of color, suppuration, assessment of the depth of the peri-implant pocket, and x-ray determination of bone loss [51].

More and more, we focus on early diagnosis of peri-implantitis and its rate of progression. The use of biomarkers can aid in the early detection of peri-implantitis [52]. Biomacromolecules, such as chemokines, MMPs, and cytokines derived from the peri-implant crevicular fluid (PICF), have been proposed as additional parameters to promote the diagnosis, prognosis, and management of peri-implant mucositis (PIM) and peri-implantitis (PI) [53]. The available biomarkers in the oral cavity are found in the saliva, the gingival crevicular fluid, the peri-implant sulcular fluid, and the mouth rinse remnant. Those fluids can be collected without being invasive, and they show great potential to detect periodontal health and periodontal disease [54].

Recent studies have shown that the level of MMP-8 is often elevated in PICF [55,56]. Elevation of MMPs is associated with the irreversible destruction of connective tissue around the implant [57] and has been attributed to a polymorphism in the promoter region of *MMP8*, which explains the different responses among individuals in the same disease category [58,59]. According to Alassiri, S. et al., low levels of MMP-8 (<20 ng/mL) in PICF are linked to periodontal health, while the upregulation is associated with an increased risk of peri-implant inflammation (Figure 3). Pathologically increased levels of MMP-8 (>20 ng/mL) can be observed by a quantitative MMP-8 chair-side device—ImplantSafe^®^, which helps to differentiate inactive from active periodontal and peri-implant sites with a sensitivity of 90% and a specificity of 70–85% [60]. Additionally, even with the high accuracy of MMP-8 in differentiating health from disease, the level of MMP-8 in saliva can be modified due to caries, smoking, and increased body mass index (BMI). This may affect accuracy in the early stages of diagnosing periodontal disease [61]. The levels of MMP-8 can also be affected by medications, such as the low doses of drugs that take part in the conventional treatment of both peri-implantitis and periodontitis (Figure 3). Their goal as an adjuvant is to modify the host inflammatory response, which usually includes tissue destruction [62]. Doxycycline (20mg) is a well-known inhibitor of MMP-8, which modulates the host immune response. Low dose of Doxycycline does not lead to a bacterial resistance or cross-resistance. For those reasons, patients on Doxycycline therapy should be examined with special attention, because of the inhibitor effect on MMP-8 [63].

As one of the major proteases in GCF, MMP-8 plays a vital role in the initiation and progression of peri-implantitis and shows a relationship with various clinical indicators. MMP-8 may also have potential application in the diagnosis and prognosis of peri-implant disease and serve as an adjunct to other related biological indicators in the diagnosis of peri-implant disease [4,32,64,65,66].

In the literature, there is a very limited number of studies focused on the *MMP8* gene variants as factors involved in peri-implantitis and implant loss. An early study by Costa-Junior et al. has explored the possible relationship between one of the functional polymorphisms in the MMP8 gene (−799 C > T, rs 11225395) and early implant failure [32]. The authors have analyzed the genotype distribution in 100 nonsmoking patients with one or more healthy implants and in 80 non-smokers that have one or more early implant failures. The results have suggested that the variant *T* allele and the *TT* genotype may be predisposing factors for early implant failure [32]. Similar results were obtained in a later case-control cross-sectional study from 2018, including 100 patients with early implant failure and 100 non-smokers with age and gender volunteers. The authors have explored the possible effect of four SNPs in four genes encoding MMPs: *MMP8* −799 C > T (rs 11225395), *MMP1* −519 A > G (rs 1144393), *MMP1* −1607 G > GG (rs 1799750) and *MMP3* −1612 5A > 6A (rs 3025058) [65]. The *T* allele and *TT* genotypes have appeared to be more frequent in test groups than in controls. A significant finding of this study is that the haplotype *T-A-GG-5A* is a risk factor, while the *C-A-G-6A* and *C-G-G-6A* are protective against implant loss [65].

## 5. MMPs and MMP-8 in Carious Lesions

Caries is considered to be a multifactorial disease, which is influenced by environmental, biological, and behavioral factors that, if present, increase the possibility of a disease occurrence [67].

Carious lesions occur when the mineral and organic matrix are dissolved. The demineralization is caused by bacterial acids and other risk factors. The degradation of the collagen is thought to be initiated by collagenolytic MMPs, more specifically, MMP-8. Studies show that patients with manifest carious lesions have higher levels of salivary MMP-8 compared to subjects without carious lesions [61].

MMPs have been found in dentine, odontoblasts, periapical tissue, and in pulp. They play an important role in maintaining homeostasis in the processes of normal tissue modeling, dentine matrix formation, and modulation of the progression of caries and secondary dentine formation. MMPs also take part in numerous extracellular pathologic conditions. Furthermore, they participate in the processes of both reversible and irreversible pulpitis, as well as in the inflammation of the periapical region [68].

MMPs take part in the processes of mineralization of the enamel and the dentin, the prevention, and the treatment of dental erosion [7]. MMPs that are found in saliva, dentinal fluid, and mineralized dentin may affect the caries of the dentin process at the early stages of demineralization. The changes in collagen suggest the participation in the lowering of the mechanical properties of the affected dentin and the reduced ability to remineralize [69].

When it comes to restorative dentistry, recent studies propose that MMPs play an important role in maintaining adequate bond strength. The endogenous dentinal MMPs are affected by the etching process and sequentially influence the bond strength [70].

The genetic factors are also explored concerning the enamel development and caries establishment. In a study with 505 Brazilian children and adolescents from 3 to 21 years of age (293 with caries and 212 caries-free), Tannure et al. have observed that the variant *G* allele of the polymorphism rs2252070 (A > G) in *MMP13* has a protective role, while none of the other studied SNPs [*MMP2* rs243865 (C > T), *MMP9* rs17576 (A > G) and *TIMP2* rs7501477 (G > T)] have demonstrated a significant association with caries experience [71]. Later, Vasconcelos et al. explored the possible predisposing role of selected polymorphisms in *MMP8* (rs17099443 C > G and rs3765620 G > A), *MMP13* (rs478927 C > T and rs2252070 C > T), and *MMP20* (rs1784418 T > C) for caries experience and developmental defects of enamel (DDE) in 216 children from the Amazon region of Brazil [72]. The main findings are that only the variant *T* allele of *MMP13* rs478927 C > T SNP is a significant risk factor for both caries experience and DDE, while the other four SNPs do not express any significant associations [72].

## 6. Conclusions

It can be concluded that MMPs indeed play an important role in the breakdown of the collagenous structure leading to the destruction of periodontal tissues and to various other pathological conditions. However, they are also essential in the physiologic process of tissue remodeling. Assessing MMP-8 is an extremely valuable tool for diagnosing and targeting this enzyme could be an efficient approach for treating periodontitis or early implant loss.

Biomarkers such as MMP-8 can assist in both staging and grading periodontitis. Future studies should focus on implementation and more efficient chair-side tests that could be used in daily practice.

## Figures and Tables

**Figure 1 biomedicines-11-01514-f001:**
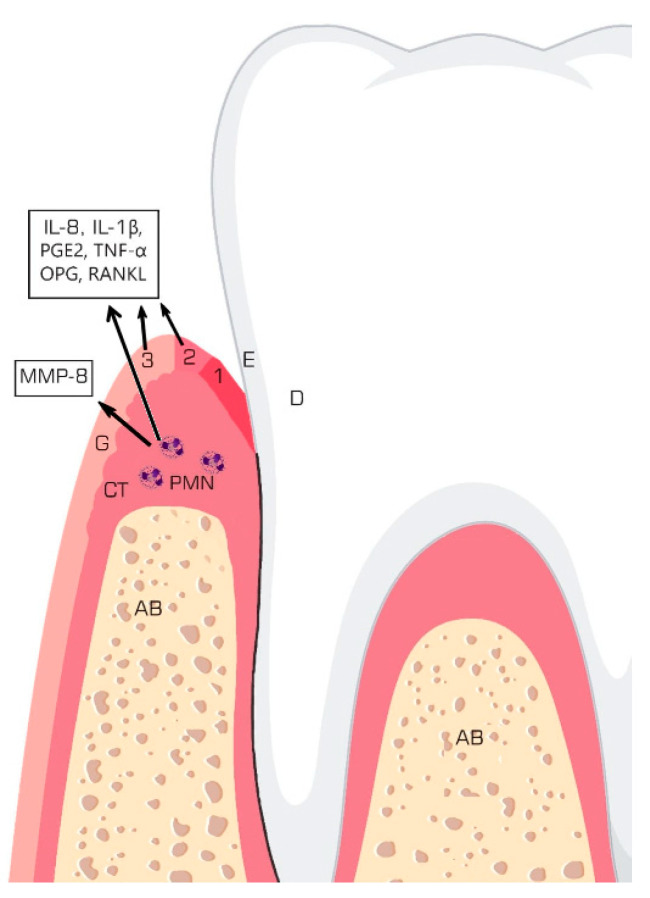
Secretion in oral fluid of MMP-8 and several regulatory molecules by activated polymorphonuclear leukocytes (PMNs) and non-immune cells (gingival epithelium) and fibroblasts in gingival connective tissue. E-enamel; D-dentin; G-gingiva; CT—gingival connective tissue: AB—alveolar bone; 1—junctional epithelium; 2—sulcular epithelium; 3—oral-gingival epithelium (modified by [3,12]).

**Figure 2 biomedicines-11-01514-f002:**
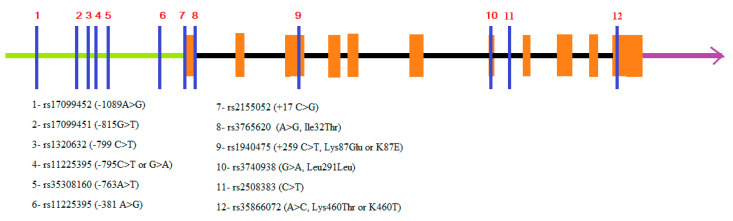
Schematic illustration of the structure of *MMP8* gene with some of the most studied functional single-nucleotide polymorphisms (SNPs) (modified [22]).

**Figure 3 biomedicines-11-01514-f003:**
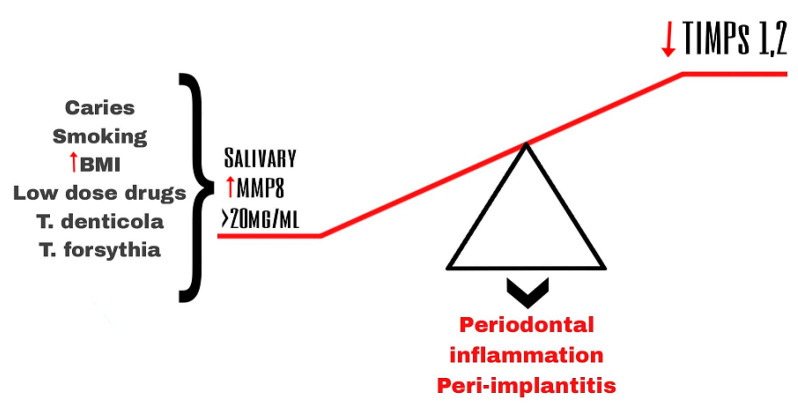
Imbalance of salivary MMP-8 and TIPMs leading to periodontal inflammation and peri-implantitis.

**Table 1 biomedicines-11-01514-t001:** Observations from studies concerning *MMP8* −799 C > T (rs11225395) +17 C/G (rs2155052) and −381A/G (rs11225395) polymorphisms in periodontitis.

Polymorphism in MMP8	Population	Disease	Patients/Controls	Observation
−799 (C > T)	Taiwan	AgP + CP	96 + 361/106	Increased risk for AgP (*p* = 0.04) and CP (*p* = 0.007) in carriers of T allele [44].
−799 (C > T)				No differences in allele (*p* = 0.06) and genotype (*p* = 0.280 distributions. No associations with particular periodontal pathogen [45]
	Czech	CP	341/278	

+17 (C > G)	Czech	CP	341/278	No differences in allele (*p* = 0.38) and genotype (*p* = 0.09) distributions. No associations with particular periodontal pathogen [45]
−799C>T/+17C > G haplotypes	Czech	CP	341/278	*-799T/+17C* haplotype is associated with 1.273-fold risk of CP (*p* = 0.038) [45]
−799 (C > T)	Turkish	GAgP	100/267	*T* allele (*p* < 0.0001) and *T* allele genotypes (*CT + TT*, *p* < 0.0001) were more common in GAgP determining 2.878-; 6.76-fold higher risk of GAgP compared to the wild *C* allele and *C*C genotype [18].
+17 (C > G)	Turkish	GAgP	100/267	No differences in allele (*p* = 0.290) and genotype (*p* = 0.581) distributions [18]
−381 (A > G)	Turkish	GAgP	100/267	*G* allele (*p* = 0.027) and *G* allele genotypes (*AG + GG*, *p* = 0.015) were less common in GAgP determining 1.5- and 2.27-fold lower risk of GAgP (OR = 0.664 and OR = 0.44) compared to the wild *A* allele and AA genotype [18]

CP—chronic periodontitis; AgP—aggressive periodontitis; GAgP—generalized aggressive periodontitis.

## Data Availability

Not applicable.

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
