# Peer review of "Matrix Metalloproteinases in Oral Health—Special Attention on MMP-8"

_biomedicines, 2023, doi:10.3390/biomedicines11061514_

Round 1

Reviewer 1 Report

Few corrections are required:

(The Authors must see my remarks)

1. Line 18 Please explain the abbreviation...

2. Line 209 Replace by ''associations''...

3. Line 318 Reduce that Section and state the main outcomes only....

Author Response

To: Jeremiah Yi

Section Managing Editor

of Biomedicines

and to: Ms. Jolie Li

Date: 17.05.2023

Dear Jeremiah Yi and Ms. Jolie Li, Please, find attached the revised version of our manuscript entitled, “Matrix metalloproteinases in oral health - special attention on MMP-8” by Atanasova T et al.,

We are very thankful to the reviewers and to the editors of Biomedicine for the comments and recommendations.

We have made corrections according to the recommendations. The English language was also corrected by native speaking person.

We hope that the manuscript now cover the high requirements of the journal Biomedicines.

Further we explain point by point all corrections we have done.

Reviewer 1:

  1. Line 18 We explained the abbreviation MMP and some other further in the text, CXCL5 (chemokine (C-X-C motif) ligand), single-nucleotide polymorphisms (SNPs), gingival crevicular fluid  (GCF),
  2. Line 209 Replace by ''associations''. - done
  3. Line 318 Reduce that Section and state the main outcomes only – we omitted the last two paragraphs at the end of the conclusions.

Thank you for your time and assistance.

Sincerely,

Prof. Tatyana Vlaykova, PhD

Medical Faculty, Trakia University, 

Dept. Chemistry and Biochemistry

11 Armeiska Str., Stara Zagora, 6000

Bulgaria

tel: ++359 888 002438

E.mail: [email protected]

Second e.mail: [email protected]

ResearcherID:          X-1463-2018

ORCID ID https://orcid.org/0000-0003-1488-8867

Scopus Author ID: 6602536162

Reviewer 2 Report

  • In this mini review the authors aimed to summarize the literature information about the functions of MMPs, as paying more attention on MMP-8. But almost all results show data from MMP8. Therefore, the title and aim should be written according to this statement. 

  •  Some conclusions are not related to the text. For instance, “Taking decision for the diagnosis of peri-implant diseases with the conventional methods can be inaccurate and sometimes harmful.” This was not an aim´s paper.

No comments

Author Response

To: Jeremiah Yi

Section Managing Editor

of Biomedicines

and to: Ms. Jolie Li

Date: 17.05.2023

Dear Jeremiah Yi and Ms. Jolie Li, Please, find attached the revised version of our manuscript entitled, “Matrix metalloproteinases in oral health - special attention on MMP-8” by Atanasova T et al.,

We are very thankful to the reviewers and to the editors of Biomedicine for the comments and recommendations.

We have made corrections according to the recommendations. The English language was also corrected by native speaking person.

We hope that the manuscript now cover the high requirements of the journal Biomedicines.

Further we explain point by point all corrections we have done.

Reviewer 2:

  1.  Some conclusions are not related to the text. - we omitted the last two paragraphs at the end of the conclusions.

Thank you for your time and assistance.

Sincerely,

Prof. Tatyana Vlaykova, PhD

Medical Faculty, Trakia University, 

Dept. Chemistry and Biochemistry

11 Armeiska Str., Stara Zagora, 6000

Bulgaria

tel: ++359 888 002438

E.mail: [email protected]

Second e.mail: [email protected]

ResearcherID:          X-1463-2018

ORCID ID https://orcid.org/0000-0003-1488-8867

Scopus Author ID: 6602536162
